# Conditional Independence Testing using Generative Adversarial Networks

**Alexis Bellot**[1,2]    **Mihaela van der Schaar**[1,2,3]

[1]University of Cambridge, [2]The Alan Turing Institute, [3]University of California Los Angeles

[abellot,mschaar]@turing.ac.uk

## Abstract

We consider the hypothesis testing problem of detecting conditional dependence, with a focus on high-dimensional feature spaces. Our contribution is a new test statistic based on samples from a generative adversarial network designed to approximate directly a conditional distribution that encodes the null hypothesis, in a manner that maximizes power (the rate of true negatives). We show that such an approach requires only that density approximation be viable in order to ensure that we control type I error (the rate of false positives); in particular, no assumptions need to be made on the form of the distributions or feature dependencies. Using synthetic simulations with high-dimensional data we demonstrate significant gains in power over competing methods. In addition, we illustrate the use of our test to discover causal markers of disease in genetic data.

## 1 Introduction

Conditional independence tests are concerned with the question of whether two variables $X$ and $Y$ behave independently of each other, after accounting for the effect of confounders $Z$. Such questions can be written as a hypothesis testing problem:

$$\mathcal{H}_0 : X \perp\!\!\!\perp Y | Z \quad \text{versus} \quad \mathcal{H}_1 : X \not\perp\!\!\!\perp Y | Z$$

Tests for this problem have recently become increasingly popular in the Machine Learning literature [19, 24, 18, 17, 6] and find natural applications in causal discovery studies in all areas of science [12, 14]. An area of research where such tests are important is genetics, where one problem is to find genomic mutations directly linked to disease for the design of personalized therapies [26, 11]. In this case, researchers have a limited number of data samples to test relationships even though they expect complex dependencies between variables and often high-dimensional confounding variables $Z$. In settings like this, existing tests may be ineffective because the accumulation of spurious correlations from a large number of variables makes it difficult to discriminate between the hypotheses. As an example the work in [16] shows empirically that kernel-based tests have rapidly decreasing power with increasing data dimensionality.

In this paper, we present a test for conditional independence that relies on a different set of assumptions that we show to be more robust for testing in high-dimensional samples $(X, Y, Z)$. In particular, we show that given only a viable approximation to a conditional distribution one can derive conditional independence tests that are approximately valid in finite samples and that have non-trivial power. Our test is based on a modification of Generative Adversarial Networks (GANs) [8] that simulates from a distribution under the assumption of conditional independence, while maintaining good power in high dimensional data. In our procedure, after training, the first step involves simulating from our network to generate data sets consistent with $\mathcal{H}_0$. We then define a test statistic to capture the $X - Y$ dependency in each sample and compute an empirical distribution which approximates the behaviour of the statistic under $\mathcal{H}_0$ and can be directly compared to the statistic observed on the real data to make a decision.

The paper is outlined as follows. In section 2, we provide an overview of conditional hypothesis testing and related work. In section 3, we provide details of our test and give our main theoretical results. Sections 4 and 5 provide experiments on synthetic and real data respectively, before concluding in section 6.

## 2 Background

We start by introducing our notation and define central notions of hypothesis testing. Throughout, we will assume the observed data consists of $n$ $i.i.d$ tuples $(X_i, Y_i, Z_i)$, defined in a potentially high-dimensional space $\mathcal{X} \times \mathcal{Y} \times \mathcal{Z}$, typically $\mathbb{R}^{d_x} \times \mathbb{R}^{d_y} \times \mathbb{R}^{d_z}$. Conditional independence tests statistics $T : \mathcal{X} \times \mathcal{Y} \times \mathcal{Z} \to \mathbb{R}$ summarize the evidence in the observational data against the hypothesis $\mathcal{H}_0 : X \perp\!\!\!\perp Y | Z$ in a real-valued scalar. Its value from observed data, compared to a defined threshold then determines a decision of whether to reject the null hypothesis $\mathcal{H}_0$ or not reject $\mathcal{H}_0$. Hypothesis tests can fail in two ways:

- Type I error: rejecting $\mathcal{H}_0$ when it is true.
- Type II error: not rejecting $\mathcal{H}_0$ when it is false.

We define the $p$-value of a test as the probability of making a type I error, and its power as the probability of correctly rejecting $H_0$ (that is 1 - Type II error). A good test requires the $p$-value to be upper-bounded by a user defined significance level $\alpha$ (typically $\alpha = 0.05$) and seeks maximum power. Testing for conditional independence is a challenging problem. Shah et al. [20] showed that no conditional independence test maintains non-trivial power while controlling type I error over any null distribution. In high dimensional samples (relative to sample size), the problem of maintaining good power is exacerbated by spurious correlations which tend to make $X$ and $Y$ appear independent (conditional on $Z$) when they are not.

### 2.1 Related work

A recent favoured line of research has characterized conditional independence in a **reproducing kernel Hilbert space** (RKHS) [24, 6]. The dependence between variables is assessed considering all moments of the joint distributions which potentially captures finer differences between them. [24] uses a measure of partial association in a RKHS to define the KCIT test with provable control on type I error asymptotically in the number of samples. Numerous extensions have also been proposed to remedy high computational costs, such as [21] that approximates the KCIT with random Fourier features making it significantly faster. Computing the limiting distribution of the test becomes harder to accurately estimate in practice [24], and different bandwidth parameters give widely divergent results with dimensionality [16], which affects power.

To avoid tests that rely on asymptotic null distributions, **sampling strategies** consider explicitly estimating the data distribution under the null assumption $\mathcal{H}_0$. Permutation-based methods [6, 17, 3, 19] follow this approach. To induce conditional independence, they select permutations of the data that preserve the marginal structure between $X$ and $Z$, and between $Y$ and $Z$. For a set of continuous conditioning variables and for sizes of the conditioning set above a few variables, the "similar" examples (in $Z$) that they seek to permute are hard to define as common notions of distance increase exponentially in magnitude with the number of variables. The approximated permutation will be inaccurate and its computational complexity will not be manageable for use in practical scenarios. As an example, [6] constructs a permutation $P$ that enforces invariance in $Z$ ($PZ \approx Z$) while [17] uses nearest neighbors to define suitable permutation sets.

We propose a different sampling strategy building on the ideas proposed by [4] that introduce the conditional randomization test (CRT). It assumes that the conditional distribution of $X$ given $Z$ is known under the null hypothesis (in our experiments we will assume it to be Gaussian for use in practice). The CRT then compares the known conditional distribution to the distribution of the observed samples of the original data using summary statistics. Instead we require a weaker assumption, namely having access to a viable approximation, and give an approximately valid test that does not depend on the dimensionality of the data or the distribution of the response $Y$; resulting in a non-parametric alternative to the CRT. [3] also expands the CRT by proposing a permutation-based approach to density estimation. **Generative adversarial networks** have been used for hypothesis

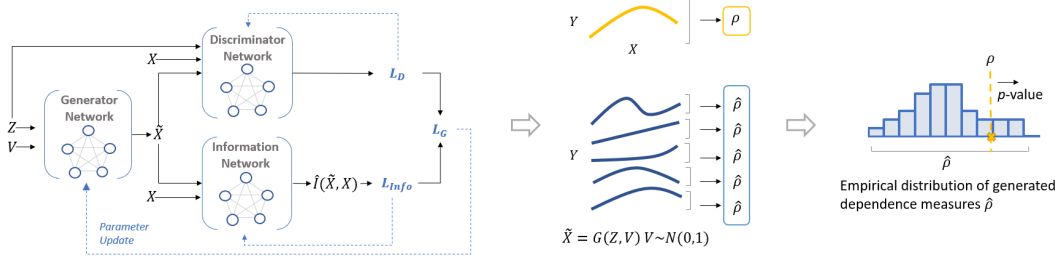

Figure 1: Illustration of conditional independence testing with the GCIT. A generator $G$ is optimized by adversarial training to estimate the conditional distribution $X|Z$ under $\mathcal{H}_0$. We then use $G$ to generate synthetic samples of $\tilde{X}$ under the estimated conditional distribution. Multiple draws are taken for each configuration $Z$ and a measure of dependence between generated $\tilde{X}$ and $Y$, $\hat{\rho}$, is computed. The sequence of synthetic $\hat{\rho}$ is subsequently compared to the original sample statistic $\rho$ to get a $p$-value.

testing in [18]. In this work, the authors use GANs to model to the data distribution and fit a classification model to discriminate between the true and estimated samples. The difference with our test is that they provide only a loose characterization of their test statistic's distribution under $\mathcal{H}_0$ using Hoeffding's inequality. As an example of how this might impact performance is that Hoeffding's inequality does not account for the variance in the data sample which biases the resulting test. A second contrast with our work is that we avoid estimating the distribution exactly but rather use the generating mechanism directly to inform our test.

## 3   Generative Conditional Independence Test

Our test for conditional independence, the GCIT (short for Generative Conditional Independence Test), compares an observed sample with a generated sample equal in distribution if and only if the null hypothesis holds. We use the following representation under $\mathcal{H}_0$,

$$Pr(X|Z,Y) = Pr(X|Z) \sim q_{\mathcal{H}_0}(X) \tag{1}$$

On the right hand side the null model preserves the dependence structure of $Pr(X, Z)$ but breaks any dependency between $X$ and $Y$. If actually there exists a direct causal link between $X$ and $Y$ then replacing $X$ with a null sample $\tilde{X} \sim q_{\mathcal{H}_0}$ is likely to break this relationship.

Sampling repeatedly $\tilde{X}$ conditioned on the observed confounders $Z$ results in an exchangeable sequence of generated triples $(\tilde{X}, Y, Z)$ and original data $(X, Y, Z)$ under $\mathcal{H}_0$. In this context, any function $\rho$ - such as a statistic $\rho : \mathcal{X} \times \mathcal{Y} \times \mathcal{Z} \to \mathbb{R}$ - chosen independently of the values of $X$ applied to the real and generated samples preserves exchangeability. Hence the sequence,

$$\rho(X, Y, Z), \rho(\tilde{X}^{(1)}, Y, Z), ..., \rho(\tilde{X}^{(M)}, Y, Z) \tag{2}$$

is exchangeable under the null hypothesis $\mathcal{H}_0$, deriving from the fact that the observed data is equally likely to have arisen from any of the above. Without loss of generality, we assume that larger values of $\rho$ are more extreme. The $p$-value of the test can be approximated by comparing the generated samples with the observed sample,

$$\sum_{m=1}^{M} \mathbf{1}\{\rho(\tilde{X}^{(m)}, Y, Z) \geq \rho(X, Y, Z)\}/M \tag{3}$$

which can be made arbitrarily close to the true probability, $\mathbb{E}_{\tilde{X} \sim q_{\mathcal{H}_0}} \mathbf{1}\{\rho(\tilde{X}, Y, Z) \geq \rho(X, Y, Z)\}$, by sampling additional features $\tilde{X}$ from $q_{\mathcal{H}_0}$. $\mathbf{1}$ is the indicator function. Figure 1 gives a graphical overview of the GCIT.

### 3.1   Generating samples from $q_{\mathcal{H}_0}$

In this section we describe a sampling algorithm that adapts generative adversarial networks [8] to generate samples $\tilde{X}$ conditional on high dimensional confounding variables $Z$. GANs provide a

powerful method for general-purpose generative modeling of datasets by designing a discriminator $D$ explicitly used as an adversary to train a generator $G$ responsible for estimating $q_{\mathcal{H}_0} := Pr(X|Z)$. Over successive iterations both functions improve based on the performance of the adversarial player.

Our implementation is based on Energy-based generative neural networks introduced in [25] which if trained optimally, can be shown to minimize a measure of divergence between probability measures that directly relates to a theoretical bound shown in this section that underlies our method. Pseudo-code for the GCIT and full details on the implementation are given in Supplement D.

**Discriminator.** We define the discriminator as a function $D_\eta : \mathcal{X} \times \mathcal{Z} \to [0,1]$ parameterized by $\eta$ that judges whether a generated sample $\tilde{X}$ from $G$ is likely to be distributed as its real counterpart $X$ or not, conditional on $Z$. We train the discriminator by gradient descent to minimize the following loss function,

$$\mathcal{L}_D := \mathbb{E}_{x \sim q_{\mathcal{H}_0}} D_\eta(x, z) + \mathbb{E}_{\tilde{v} \sim p(v)} \left(1 - D_\eta(G_\phi(v, z), z)\right) \tag{4}$$

where $G_\phi(z, v), v \sim p(v)$ is a synthetic sample from the generator (described below) and $x \sim q_{\mathcal{H}_0}$ is a sample from the data distribution under $\mathcal{H}_0$. Note that in contrast to [25] we set the image of $D$ to lie in $(0,1)$ and include conditional data generation.

**Generator.** The generator, $G$, takes (realizations of) $Z$ and a noise variable, $V$, as inputs and returns $\tilde{X}$, a sample from an estimated distribution $X|Z$. Formally, we define $G : \mathcal{Z} \times [0,1]^d \to \mathcal{X}$ to be a measurable function (specifically a neural network) parameterized by $\phi$, and $V$ to be $d$-dimensional noise variable (independent of all other variables). For the remainder of the paper, let us denote $\tilde{x} \sim \hat{q}_{\mathcal{H}_0}$ the generated sample under the model distribution implicitly defined by $\hat{x} = G_\phi(v, z), v \sim p(v)$. In opposition to the discriminator, $G$ is trained to minimize

$$\mathcal{L}_G(D) := \mathbb{E}_{\tilde{x} \sim \hat{q}_{\mathcal{H}_0}} D_\eta(\tilde{x}, z) - \mathbb{E}_{x \sim q_{\mathcal{H}_0}} D_\eta(x, z) \tag{5}$$

We estimate the expectations empirically from real and generated samples.

## 3.2 Validity of the GCIT

The following result ensures that our sampling mechanism leads to a valid test for the null hypothesis of conditional independence.

**Proposition 1** (Exchangeability) *Under the assumption that $X \perp\!\!\!\perp Y | Z$, any sequence of statistics $(\rho_i)_{i=1}^M$ functions of the generated triples $(\tilde{X}^{(m)}, Y, Z)_{m=1}^M$ is exchangeable.* $\qquad\square$

*Proof.* All proofs are given in Supplement C.

Generating conditionally independent samples with a neural network preserves exchangeability of input samples and thus leads to a valid $p$-value, defined in eq. (3), for the hypothesis of conditional independence. Under the assumption that the conditional distribution $q_{\mathcal{H}_0}$ can be estimated exactly, this implies that we maintain an exact control of the type I error in finite samples. In practice however, limited amounts of data and noise will prevent us from learning the conditional distribution exactly.

In such circumstances we show below that the excess type I error - that is the proportion of false negatives reported above a specified tolerated level $\alpha$ - is bounded by the loss function $\mathcal{L}_G$; which, moreover, can be made arbitrarily close to 0 for a generator with sufficient capacity. We give this second result as a corollary of the GAN's convergence properties in Supplement C.

**Theorem 1** *An optimal discriminator $D^*$ minimizing $\mathcal{L}_D$ exists; and, for any statistic $\hat{\rho} = \rho(X, Y, Z)$, the excess type I error over a desired level $\alpha$ is bounded by $\mathcal{L}_G(D^*)$,*

$$Pr(\hat{\rho} > c_\alpha | \mathcal{H}_0) - \alpha \leq \mathcal{L}_G(D^*) \tag{6}$$

*where $c_\alpha := \inf\{c \in \mathbb{R} : Pr(\hat{\rho} > c) \leq \alpha\}$ is the critical value on the test's distribution and $Pr(\hat{\rho} > c_\alpha | \mathcal{H}_0)$ is the probability of making a type I error.* $\qquad\square$

Theorem 1 shows that the GCIT has an increase in type I error dependent only on the quality of our conditional density approximation, given by the loss function with respect to the generator, even in the worst-case under *any* statistic $\rho$. For reasonable choices of $\rho$, robust to errors in the estimation

of the conditional distribution, this bound is expected to be tighter. The *key* assumption to ensure control of the type I error, and therefore to ensure the validity of the GCIT, thus rests solely on our ability to find a viable approximation to the conditional distribution of $X|Z$. The capacity of deep neural networks and their success in estimating heterogeneous conditional distributions even in high-dimensional samples make this a reasonable assumption, and the GCIT applicable in a large number of scenarios previously unexplored.

## 3.3 Maximizing power

For a fixed sample size, conditional dependence $\mathcal{H}_1 : X \not\perp Y|Z$, is increasingly difficult to detect with larger conditioning sets ($Z$) as spurious correlations due to sample size make $X$ and $Y$ appear independent. To maximize power it will be desirable that differences between generated samples $\tilde{X}$ (under the model $Pr(X|Z)$) and observed samples $X$ (distributed according to $Pr(X|Z,Y)$) be as apparent as possible. In order to achieve this we will encourage $\hat{X}$ and $X$ to have low mutual information because irrespective of dimensionality, mutual information between distributions in the null and alternative relates directly to the hardness of hypothesis testing problems, which can be seen for example via Fano's inequality (section 2.11 in [23]). To do so, we investigate the use of the information network proposed in [2] and used in the context of feature selection in [10]. [2] propose a neural architecture and training procedure for estimating the mutual information between two random variables. We approximate the mutual information with a neural network $T_\theta : \mathcal{X} \times \mathcal{X} \to \mathbb{R}$, parameterized by $\theta$, with the following objective function (to be maximized),

$$\mathcal{L}_{Info} := \sup_{\theta} \mathbb{E}_{p_{x,\tilde{x}}^{(n)}}[T_\theta] - \log \mathbb{E}_{p_x^{(n)} \times p_{\tilde{x}}^{(n)}}[\exp(T_\theta)] \tag{7}$$

We estimate $T_\theta$ in alternation with the discriminator and generator given samples from the generator in every iteration. We modify the loss function for the generator to include the mutual information and perform gradient descent to optimize the generator on the following objective,

$$\mathcal{L}_G(D) + \lambda \mathcal{L}_{Info} \tag{8}$$

$\lambda > 0$ is a hyperparameter controlling the influence of the information network. This additional term ($\lambda \mathcal{L}_{Info}$) encourages the generation of samples $\tilde{X}$ as independent as possible from the observed variables $X$ such that the resulting differences (between $\tilde{X}$ and $X$) are truly a consequence of the *direct* dependence between $X$ and $Y$ rather than spurious correlations with confounders $Z$.

To provide some further intuition, one can see why generating data different than the sample observed in the alternative $\mathcal{H}_1$ might be beneficial by considering the following bound (proven in Supplement C),

$$\text{Type I error} + \text{Type II error} \geq 1 - \delta_{TV}(\hat{q}_{\mathcal{H}_0}, q_{\mathcal{H}_1}) \tag{9}$$

where $\hat{q}_{\mathcal{H}_0}$ is the estimated null distribution with the GCIT, $q_{\mathcal{H}_1}$ is the distribution under $\mathcal{H}_1$ and where $\delta_{TV}$ is the total variation distance between probability measures. This result suggests that when emphasizing the differences between the estimated samples and true samples from $\mathcal{H}_1$, which increases the total variation, can improve the overall performance profile of our test by reducing a lower bound on type I and type II errors.

**Remark.** The GCIT aims at generating samples whose conditional distribution matches the distribution of its real counterparts, but can be independent otherwise. It is that gap that the power maximizing procedure intends to exploit. In practice, there will be a trade-off between the objectives of the discriminator and information network but we found that setting $\lambda = 10$ in our experiments achieved good performance. It should be noted also that hyperparameter selection cannot be performed using cross-validation as we do not have access to ground truth and so the hyperparameters must typically be fixed a priori. However, we can consider artificially inducing conditional independence ($X \perp\!\!\!\perp Y|Z$) (by permuting variables $X$ and $Y$ such as to preserve the marginal dependence in $(X, Z)$ and $(Y, Z)$) and choose hyperparameters that best control for type I error. We explore this further in Supplement A and test configurations of $\lambda$ with synthetic data in section 4.2.

## 3.4 Choice of statistic $\rho$

The bound on the type I error given in Theorem 1 holds for any choice of statistic $\rho$ as it depends solely on the conditional distribution estimation. For choices of $\rho$ less sensitive to spurious differences

between generated and true samples when the null $\mathcal{H}_0$ holds, the type I error is expected to be below this bound. We experimented with various dependence measures (between two samples) as choices for $\rho$. We consider the Maximum Mean Discrepancy [9], Pearson's correlation coefficient, the distance correlation (which measures both linear and nonlinear association, in contrast to Pearson's correlation), the Kolmogorov-Smirnov distance between two samples and the randomized dependence coefficient [13]. In our experiments we use the distance correlation and analyze performance using all other measures in Supplement A.

## 4 Synthetic data example

In this section we analyse the performance of the GCIT[1] in a controlled fashion with synthetic data against a wide range of competing algorithms, illustrating the effects of different components of our method. We consider the CRT [4] with pre-specified Gaussian sampling distribution, whose parameters are estimated from data; the kernel-methods KCIT [24] and RCoT [21] with bandwith parameter estimated with the median of all pairwise distances between $X$ and $Y$, a common choice in the literature; and the CCIT [19], which does not make prior assumptions on data distributions but was also not specifically designed for high-dimensional data.

When testing at level $\alpha$, type I error should be as close as possible to $\alpha$ even though this might not be the case because of violated assumptions or approximations. An important consideration in our discussion of power as we increase the dimensionality of $Z$, is the choice of alternatives $\mathcal{H}_1$. For instance, if the strength of the dependency between $X$ and $Y$ increases, the hypothesis testing problem will be made artificially easier and bias our conclusions with regards to data dimensionality, as observed also in [16]. In every synthetic experiment, we maintain the mutual information between $X$ and $Y$ approximately constant by first generating data and second estimating the mutual information before deciding to draw a new dataset, if the mutual information disagrees with the previous draw, or otherwise proceed with testing. We estimate the mutual information with a Gaussian approximation, $MI(X, Y) = -\frac{1}{2} \log(1 - \hat{\rho}^2)$, where $\hat{\rho}$ is the linear correlation between $X$ and $Y$.

### 4.1 Setup

We generate synthetic data according to the "post non-linear noise model" similarly to [24, 6, 21] that defines $(X, Y, Z)$ under $\mathcal{H}_0$ and $\mathcal{H}_1$ as follows,

$$\mathcal{H}_0: \quad X = f(A_f Z + \epsilon_f), \quad Y = g(A_g Z + \epsilon_g) \tag{10}$$

$$\mathcal{H}_1: \quad Y = h(A_h Z + \alpha X + \epsilon_h) \tag{11}$$

The matrix dimensions of $A_{(.)}$ are such that $X$ and $Y$ are univariate, matrix entries as well as parameter $\alpha$ are generated at random in the interval $[0, 1]$, and lastly, the noise variables $\epsilon_{(.)}$ are 0 on average with variance 0.025. The distributions of $X$, $Y$ and $\epsilon$, and the complexity of dependencies via $f, g$ and $g$ will be tuned carefully to make performance comparisons in three settings:

**(1) Multivariate Gaussian**
We set $f, g$ and $h$ to be the identity functions which induces linear dependencies, $Z \sim \mathcal{N}(0, \sigma^2)$, and $X \sim \mathcal{N}(0, \sigma^2)$ under $\mathcal{H}_1$ which results in jointly Gaussian data under the null and the alternative. Such a setting matches the assumptions of all methods and the interest of this study will be to provide a baseline for more complex scenarios.

**(2) Multivariate Laplace**
Kernel choice has a large impact on power, as we demonstrate in this setting. In this case, we set $f, g$ and $h$ as before but use a Laplace distribution to generate $Z$ and $X$. The RBF kernel in this case overestimates the "smoothness" of the data. This study highlights the robustness of the GCIT in comparison to kernel-based methods which is important since hyperparameters cannot be tuned by cross-validation.

**(3) Arbitrary distributions**
We set $f, g$ and $h$ to be randomly sampled from $\{x^3, \tanh x, \exp(-x)\}$, resulting in more complex distributions and variable dependencies. Here $Z \sim \mathcal{N}(0, \sigma^2)$, and $X \sim \mathcal{N}(0, \sigma^2)$ under $\mathcal{H}_1$. This

`https://bitbucket.org/mvdschaar/mlforhealthlabpub/src/master/alg/gcit/`.

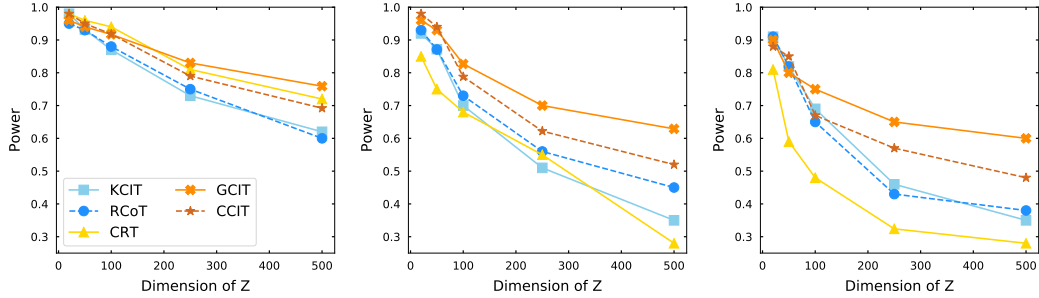

Figure 2: Power results of the synthetic simulations. (Higher better). **Left panel:** (1) Multivariate Gaussian, **Middle panel:** (2) Multivariate Laplace, **Right panel:** (3) Arbitrary distributions.

is our most general setting which most faithfully resembles the complexities we can expect in real applications.

**Results:** Power as a function of the dimensionality of $Z$ is shown in Figure 2. Each point on the curves is computed by taking averages over 1000 random experiments with sample size equal to 500 examples. The results from scenario **(1)** are consistent with our expectations; all methods perform comparably, the CRT and kernel-based methods achieving high power in lower dimensions while slightly under-performing in higher dimensions. In scenario **(2)** and **(3)**, the failure of the CRT and kernel-based methods is apparent while the GCIT maintains high power, even with increasing dimensionality, which demonstrates the robustness of our sampling mechanism to arbitrary complex data distributions. The CCIT outperforms kernel-based methods in these cases also. An important contrast of the GCIT with respect to the CCIT is our addition of the information network, which we argue contributes to the higher power observed across all experiments. We analyze this empirically below.

Figure 2 in Supplement B shows that type I error is approximately controlled at a level $\alpha$ for all methods. Observe also that even though the GCIT requires training a new GAN in every iteration, in Figure 3 Supplement B we show empirically that running times for the GCIT scale much better with dimensionality and sample size in comparison with the best benchmark, the CCIT: its running times are prohibitive in practice with more than 1000 samples or 500 dimensions in $Z$, with each test taking over $600s$ versus $60s$ for the GCIT.

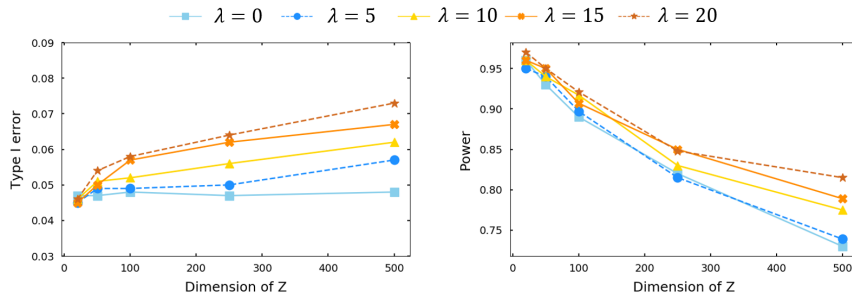

Figure 3: Type I error and power for different values of $\lambda$.

## 4.2 Source of gain: consequences of the information network

The information network aims to encourage maximum power in high-dimensional data. We control for its influence by varying $\lambda$ in the loss function of the GCIT given in eq. (8). Higher values of $\lambda$ encourage the generation of independent samples which improves power even though it might decrease the accuracy of the density approximation in the GAN optimization when the null in fact holds. We notice this trade-off between power and type I error for higher values of $\lambda$ in Figure 3. The underlying data was generated from setting **(1)**, each curve in the two panels corresponds to

a different value of $\lambda$. Lastly, we computed the lower-bound from GCIT generated samples and observed samples (by numerical integration) in eq. (9) to conclude that higher values of $\lambda$ did decrease the lower bound, as expected.

## 5 Genetic data example

There is compelling evidence that the likelihood of a patient's cancer responding to treatment can be strongly influenced by alterations in the cancer genome [7]. We study the response of cancer cell lines to an anti-cancer drug where the problem is to distinguish between genetic mutations that influence directly the cancer cell line response from those that are not directly relevant [1, 22]. We use the subset of the CCLE data [1] relating to the drug PLX4720; it contains 474 cancer cell lines described by 466 genetic mutations. More details on the data can be found in Supplement E.

| | AGREEMENT | | | | | | | DISAGREEMENT | | |
| | BRAF.V600E | BRAF.MC | HIP1 | CDC42BPA | THBS3 | DNMT1 | PRKD1 | FLT3 | PIP5K1A | MAP3K5 |
|---|---|---|---|---|---|---|---|---|---|---|
| EN | 1 | 3 | 4 | 7 | 8 | 9 | 10 | 5 | 19 | 78 |
| RF | 1 | 2 | 3 | 8 | 34 | 28 | 18 | 14 | 7 | 9 |
| CRT | <0.001 | <0.001 | 0.008 | 0.009 | 0.017 | 0.022 | 0.002 | 0.017 | 0.024 | 0.012 |
| GCIT | <0.001 | <0.001 | 0.008 | 0.050 | 0.013 | 0.020 | 0.002 | 0.521 | 0.001 | <0.001 |

Figure 4: Genetic experiment results. Each cell gives the $p$-value or importance rank (where appropriate) indicating the dependency between a mutation and drug response.

Evaluating conditional independence relations from real data is difficult as we do not have access to the ground truth causal links. Instead we give our results in comparison to those of [1], who proceeded by reporting discriminative features returned by the parameter values of a fitted elastic net regression model (EN). This is common practice in genetic studies, see for example also [7]. In addition, we compare with the rank of each feature given by a random forest model importance scores (RF) and the $p$-value assigned by the CRT. The results for 10 selected mutations can be found in Figure 4. The first two rows give ranks of heuristic methods and the last two rows give $p$-values of conditional independence tests. We distinguish between the mutations where all methods agree (in the leftmost columns), and the mutations where not all methods agree (in the rightmost columns).

The mutations on genes PIP5K1A and MAP3K5 are recognized to be discriminative by the random forest model (high rank) and the GCIT (low $p$-value), which highlights the significance of the GCIT for conditional independence testing, suggesting that non-linear dependencies occur which are not captured by the elastic net or the CRT. For further evaluation, in this case we were able to cross-reference with a previous study to find evidence of the PIP5K1A gene to have a differential response on cancer cell lines when PLX4720 is applied [22]. The MAP3K5 gene has not previously been reported in the literature as being directly linked to the PLX4720 drug response, however [15] did find a proliferation of these gene mutations to be of BRAF *type* in cancer patients. This is interesting because PLX4720 is precisely designed as a BRAF inhibitor, and thus we would expect it to have an impact also on MAP3K5 mutations of the BRAF type. FLT3 is an interesting gene, found to be dependent on cancer response by the EN, RF and CRT, but not by the GCIT. This finding by the GCIT was confirmed however by a posterior genetic study [5] that established no link between cancer response and FLT3 mutations in the presence of PLX4720. Such results encourage us to believe that the GCIT is able to better detect dependence for these problems.

## 6 Conclusions and future perspectives

We propose a generative approach to conditional independence testing using generative adversarial networks. We show this approach results in an approximately valid test for an arbitrary data distribution irrespective of the number of variables observed. We have demonstrated through simulated data significant gains in statistical power, and we illustrated the application of our method to discover genetic markers for cancer drug response on real high-dimensional data.

From a practical perspective, algorithms based on other generative models can be constructed based on our proposed procedure that may be more adequate for different data modalities. In a general sense, this work opens the door to principled statistical testing with more heterogeneous data, and expands our ability to reason and test variable relationships in more challenging scenarios.

## 7 Acknowledgements

We thank the anonymous reviewers for valuable feedback. This work was supported by the Alan Turing Institute under the EPSRC grant EP/N510129/1, the ONR and the NSF grants number 1462245 and number 1533983.

## Footnotes

[1]An implementation of our test and tutorial are available at

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
