[Supplementary Material]

# Supplementary Material: Conditional Independence Testing using Generative Adversarial Networks

**Alexis Bellot**[1,2]     **Mihaela van der Schaar**[1,2,3]

[1]University of Cambridge, [2]The Alan Turing Institute, [3]University of California Los Angeles

[abellot,mschaar]@turing.ac.uk

This Supplement contains further experiments relating to hyperparameters and computational complexity, the proofs of all statements made in the main body of this paper, implementation details for all methods compared and finally, further details on the genetic data used for illustrating the use of our conditional independence test.

## A. Discussion on hyperparameter choice

As ground truth information on variable relationships is rarely available, choosing hyperparameters is challenging. In this section we analyze the GCIT's performance as a function of hyperparameter configurations of the GCIT and discuss an approximate procedure to guide hyperparameter optimization on a validation set.

**Choice of statistic $\rho$**

We start by analyzing potential choices for the summary statistic $\rho : (\mathcal{X} \times \mathcal{Y} \times \mathcal{Z}) \times (\mathcal{X} \times \mathcal{Y} \times \mathcal{Z}) \to \mathbb{R}$ that summarizes the generated and observed samples into a real-valued scalar. Different choices for $\rho$ encode in more or less detail the distributional differences in samples and thus we can expect them to influence the resulting performance of the test. We considered the following distance and correlation measures between two samples:

- The Maximum mean discrepancy (MMD) is defined as the largest difference between the mean function values on two samples in a reproducing kernel Hilbert space. When MMD is large, the samples are likely from different distributions. For a kernel $k : \mathcal{X} \times \mathcal{X} \to \mathbb{R}$, a consistent empirical estimate of the MMD is given by [7],

$$\hat{\rho}(\mathbf{x}, \mathbf{y}) := \frac{1}{n^2} \sum_{i,j} k(x_i, x_j) + \frac{1}{n^2} \sum_{i,j} k(y_i, y_j) - \frac{2}{n^2} \sum_{i,j} k(x_i, y_j)$$

- The Pearson's correlation coefficient (PCC) is a measure of linear correlation between two variables. It is defined as,

$$\hat{\rho}(\mathbf{x}, \mathbf{y}) := \frac{\sum_i (x_i - \bar{x})(y_i - \bar{y})}{\sqrt{\sum_i (x_i - \bar{x})}\sqrt{\sum_i (y_i - \bar{y})}}$$

- The distance correlation (DC) measures both linear and nonlinear association between two random variables or random vectors. It is defined as,

$$\hat{\rho}(\mathbf{x}, \mathbf{y}) := \frac{dCov(\mathbf{x}, \mathbf{y})}{\sqrt{dVar(\mathbf{x})dVar(\mathbf{y})}}$$

- The Kolmogorov-Smirnov statistic (KS) is defined as the sup-norm between cumulative distribution functions of two samples as follows,

$$\hat{\rho}(\mathbf{x}, \mathbf{y}) := \sup_w |F_x^{(n)}(w) - F_y^{(n)}(w)|$$

where $F_x^{(n)}$ and $F_y^{(n)}$ are the empirical distribution functions of $X$ and $Y$ samples respectively.

- The Randomized Dependence Coefficient (RDC) measures the dependence between random samples $X$ and $Y$ as the largest canonical correlation between $k$ randomly chosen nonlinear projections of their copula transformations. It is formally defined an analyzed in [9].

$$\hat{\rho}(\mathbf{x}, \mathbf{y}) := \sup_{\alpha, \beta} PCC(\alpha^T \Phi_{\mathbf{x}}, \beta^T \Phi_{\mathbf{y}})$$

where $PCC$ is Pearson's correlation coefficient and $\Phi$ are nonlinear random projections, such as sine or cosine projections. See [9] for more details.

We tested the above metrics with simulated data under setting **(3)** described in the main paper. Type I error and power results for the GCIT implemented with each one of the above choices for $\rho$ are given in Figure 1. Finer differences are given by the MMD, the RDC or the DC that all consider non-linear relationships between variables; we see in the power computations in the right column that this results in higher power of the GCIT since the underlying data generating mechanism in non-linear. However, these statistics will also encode spurious differences between samples when the null $\mathcal{H}_0$ is in fact true, resulting in higher type I error. We can see this behaviour in the type I error results on the panels in the left column. The PCC, for example, that encodes only linear differences between samples is more robust to type I error.

Figure 1: Power and type I error results for different choices of $\rho$.

**Remark on the robustness of the GCIT for practical applications.** *Our test does depend to some extent on the hyperparameter configurations of both $\lambda$ and $\rho$. Recall that no ground truth is available to optimize hyperparameters using conventional methods, but we argue that the following procedure can be used to guide hyperparameter selection. We consider artificially inducing conditional independence $(X \perp\!\!\!\perp Y | Z)$ by permuting variables $X$ and $Y$ such as to preserve the marginal dependence in $(X, Z)$ and $(Y, Z)$, as in [6] (further details are also described in our related work section). On this data, a well calibrated test is expected to produce uniformly distributed $p$-values, i.e. the empirical distribution of $p$-values should be approximately uniform. Our recommendation would be to choose GCIT's hyperparameters with lowest Kolmogorov-Smirnov statistic in comparison to the uniform distribution. This ensures the resulting test produces "well-behaved" $p$-values and thus prevents to some extent $p$-value cheating. We will discuss this further in the revised manuscript, thank you for raising this point.*

## B. Further experiments and complexity analysis

In this section we present results on the type I error of the GCIT and all baseline algorithms for the synthetic simulations considered in the main body of this paper, and analyze computational complexity as a function of sample size and data dimensionality.

### Type I error versus dimensionality of $Z$

Next we show in Figure 2 type I error as a function of dimensionality of $Z$ for each one of the three synthetic simulations considered in the main body of this paper. We observe that in all cases, type I error is approximately controlled at the chosen level $\alpha = 0.05$ when the distributional assumptions underlying each method holds. This is not the case otherwise, the CRT fails to control type I error in the non-linear setting when a Gaussian approximation to the joint distribution of the variables is not appropriate.

Figure 2: Type I error results for the synthetic simulations.

### Computational complexity analysis

We give in Figure 3 the run times in seconds of all algorithms for a single conditional independence test for data generated under setting **(1)** in the main body of this paper. We vary both the number of samples (fixing the dimension of $Z$ to 100) and the dimensionality of $Z$ (fixing the sample size to 100). The GCIT scales very well with both sample size and conditioning set size, even if each iteration requires training a new GAN. In contrast, the running times of KCIT for sample sizes above 1000 and those of CCIT in higher dimensional samples are prohibitive in practice.

Figure 3: Running times in seconds as a function of sample size and dimension of $Z$.

### Experiments as a function of sample size and stability of generated p-values

We investigate the influence of sample size on the three leftmost panels of Figure 4. The GCIT, as well as most competing tests, have slightly higher type I error in low sample sizes but control type I error successfully with 500 samples or more. In terms of power, our experiments show that we

can expect the GCIT to outperform competing tests with 500 samples or more (for dimension of $Z = 100$). Next, we investigate the stability of $p$-values as a function of sample size; the variance of the empirical $p$-values quickly drops to 0. This means that for say 500 samples, we can expect the $p$-values of two independently trained GCITs to be within 0.005 of each other with approximately 95% confidence. The last panel on the right illustrates how quickly the $p$-value approximation (eq. 3 in the main body of this paper) converges to its population quantity as a function of the number of samples used to compute the approximation i.e. $M$ in eq. 3. The convergence should be at least of order $M^{-1/2}$ by the central limit theorem.

Figure 4: **Leftmost and middle-left panel**: Type I error and power as a function of sample size for data generated under scenario (3) with dimensionality of $Z$ set to 100; **Middle-right panel**: Empirical $p$-value variance of the GCIT as a function of sample size (computed by generating 100 $p$-values for each GAN trained on data with the specified size); **Rightmost panel**: Illustration of the convergence of the GCIT's $p$-values as a function of generated samples.

## C. Theoretical results

**Proof of Proposition 1.** A sequence of random variables is said to be exchangeable if its distribution is invariant under variable permutations. We make use of the "representation theorem" for exchangeable sequences of random variables, first stated by de Finetti and extended by Diaconis and Freedman for finite sequences [4, 5]. They show that every sequence of conditionally $i.i.d.$ random variables can be considered as a sequence of exchangeable random variables. With our definition of the generator we start from $i.i.d.$ sequence of noise random variables $\{V_m\}_{m=1}^M$ and define, for every $m$, $\tilde{X}^{(m)} = \phi(Z, V_m)$ where $Z$ is a random variable independent of $V_m$ and $\phi$ is a measurable function, such as a neural network in our case. By construction, the resulting random sequence of data sets $(\tilde{X}^{(m)}, Y, Z)_{m=1}^M$ is exchangeable and therefore also the sequence of statistics $(\rho_i)_{i=1}^M$ (measurable functions of $(\tilde{X}^{(m)}, Y, Z)_{m=1}^M$) is exchangeable. $\square$

The theoretical results that follow are proven only for the version of the generator loss given in equation (5) in the main body of this paper, $\mathcal{L}_G(D) := \mathbb{E}_{\tilde{x} \sim \hat{q}_{\mathcal{H}_0}} D_\eta(\tilde{x}) - \mathbb{E}_{x \sim q_{\mathcal{H}_0}} D_\eta(x)$ though we do believe that the theorem holds more generally with the addition of the power maximizing procedure - this is backed up by our empirical results demonstrating Type I error control while using the power maximizing procedure. We prove the bound on the excess Type I error in two parts. First we show in the following lemma that an optimal discriminator exists, and second we prove the bound on the Type I error.

**Lemma 1** *An optimal discriminator $D^*$ minimizing $\mathcal{L}_D := \mathbb{E}_{x \sim q_{\mathcal{H}_0}} D(x) + \mathbb{E}_{x \sim \hat{q}_{\mathcal{H}_0}} (1 - D(x))$ over all measurable functions $D$ such that $D \in (0, 1)$ exists and it is given by $D^* = \frac{1}{2} sign \left( q_{\mathcal{H}_0} - \hat{q}_{\mathcal{H}_0} \right) + \frac{1}{2}$.*

*Proof.* To see this note first that the $sign(x)$ function is defined as $+1$ or $-1$ depending on the sign of $x$. Then,

$$\mathcal{L}_{D^*} = 1 + \mathbb{E}_{x \sim q_{\mathcal{H}_0}} D^*(x) - \mathbb{E}_{x \sim \hat{q}_{\mathcal{H}_0}} D^*(x) \tag{1}$$

$$= 1 + \int_{\mathcal{X}} q_{\mathcal{H}_0}(x) \frac{1}{2} sign \left( q_{\mathcal{H}_0} - \hat{q}_{\mathcal{H}_0} \right) dx - \int_{\mathcal{X}} \hat{q}_{\mathcal{H}_0}(x) \frac{1}{2} sign \left( q_{\mathcal{H}_0} - \hat{q}_{\mathcal{H}_0} \right) dx \tag{2}$$

$$= 1 - \frac{1}{2} \int_{x : \hat{q}_{\mathcal{H}_0}(x) - q_{\mathcal{H}_0}(x) > 0} q_{\mathcal{H}_0}(x) - \hat{q}_{\mathcal{H}_0}(x) dx - \frac{1}{2} \int_{x : \hat{q}_{\mathcal{H}_0}(x) - q_{\mathcal{H}_0}(x) < 0} \hat{q}_{\mathcal{H}_0}(x) - q_{\mathcal{H}_0}(x) dx \tag{3}$$

$$= 1 - \frac{1}{2} \int_{\mathcal{X}} |\hat{q}_{\mathcal{H}_0}(x) - q_{\mathcal{H}_0}(x)| dx \tag{4}$$

$$= 1 - \frac{1}{2} \sup_{||D||_\infty \leq 1} \mathbb{E}_{x \sim \hat{q}_{\mathcal{H}_0}} D(x) - \mathbb{E}_{x \sim q_{\mathcal{H}_0}} D(x) \tag{5}$$

$$= 1 - \sup_{||D||_\infty \leq 1/2} \mathbb{E}_{x \sim \hat{q}_{\mathcal{H}_0}} D(x) - \mathbb{E}_{x \sim q_{\mathcal{H}_0}} D(x) \tag{6}$$

$$= 1 - \sup_{0 \leq D \leq 1} \mathbb{E}_{x \sim \hat{q}_{\mathcal{H}_0}} D(x) - \mathbb{E}_{x \sim q_{\mathcal{H}_0}} D(x) \tag{7}$$

$$= \inf_{0 \leq D \leq 1} \mathbb{E}_{x \sim q_{\mathcal{H}_0}} D(x) + \mathbb{E}_{x \sim \hat{q}_{\mathcal{H}_0}} (1 - D(x)) \tag{8}$$

$$\leq \mathcal{L}_D \tag{9}$$

for any $D$ in the mentioned space and with equality if and only if $D = D^*$. Eq (5) follows from the Kantorovich-Rubinstein dual representation for general $f$ divergences, proven for example in [13]. $\square$

**Corollary 1** For a generator $G$ with infinite capacity converging to the true conditional distribution $q_{\mathcal{H}_0}(x)$, $\mathcal{L}_G(D^*)$ attains its minimum value of 0.

**Proof** By setting $D^*$ in the loss of the generator $\mathcal{L}_G$ we observe that,

$$\mathcal{L}_G(D^*) = 1 - \mathcal{L}_{D^*} \tag{10}$$

$$= \frac{1}{2} \int_{\mathcal{X}} |\hat{q}_{\mathcal{H}_0}(x) - q_{\mathcal{H}_0}(x)| dx \tag{11}$$

Hence, for a generator with infinite capacity converging to the true conditional distribution $q_{\mathcal{H}_0}(x)$, the last term is 0 which implies $\mathcal{L}_G(D^*) = 0$. $\square$

**Proof of Theorem 1**. Our derivation is similar to [3]. By definition the statistic $\hat{\rho}$ results in a $p$-value $p < \alpha$ if and only if the observed variable $x$ is contained in the set $A_\alpha := \{x : \sum_{m=1}^M \mathbf{1}\{\rho(x^{(m)}, y, z) \geq \rho(x, y, z)\}/M < \alpha\}$. Consider generating a new sample $\tilde{x}$ from the generator $G$ under the estimated conditional distribution and let $x \sim q_{\mathcal{H}_0}$ be sampled from the true conditional. Then it holds that,

$$\mathcal{L}_G(D^*) = \mathbb{E}_{x \sim \hat{q}_{\mathcal{H}_0}} D^*(x) - \mathbb{E}_{x \sim q_{\mathcal{H}_0}} D^*(x) \tag{12}$$

$$= \int_{\mathcal{X}} |q_{\mathcal{H}_0}(x) - \hat{q}_{\mathcal{H}_0}(x)| dx \tag{13}$$

$$= \sup_A |q_{\mathcal{H}_0}(A) - \hat{q}_{\mathcal{H}_0}(A)| \tag{14}$$

$$\geq Pr(x \in A_\alpha) - Pr(\tilde{x} \in A_\alpha) \tag{15}$$

$$\geq Pr(\hat{\rho} > c_\alpha | \mathcal{H}_0) - \alpha \tag{16}$$

where by expanding the expectations, eq. (13) follows from similar arguments to those presented in Lemma 1. Next eq. (14) follows from a well known equivalent representation of the total variation divergence between probability measures, proven for example in Proposition 4.2, page 48 of [8]. Eq. (15) follows by standard properties of the supremum operator. Finally we arrive at eq. (16) given the fact that, by definition of $A_\alpha$, $Pr(x \in A_\alpha) = Pr(\hat{\rho} > c_\alpha | \mathcal{H}_0)$ and, since given $y$ and $z$ the set $(\tilde{x}, x^{(1)}, ..., x^{(M)})$ is conditionally independent and therefore exchangeable, we have that $Pr(\tilde{x} \in A_\alpha) \leq \alpha$. $\square$

**Proof of equation** (9). Let $q_{\mathcal{H}_0}$ and $q_{\mathcal{H}_1}$ be the true conditional distributions of $X|Z$ under the null hypothesis $H_0 : X \perp\!\!\!\perp Y|Z$ and its alternative $H_1 : X \not\perp\!\!\!\perp Y|Z$ respectively. Denote by $A$ the event that samples $x$ result in a $p$-value below the level $\alpha$. Then,

$$\text{Type I error} + \text{Type II error} = q_{\mathcal{H}_0}(A) + q_{\mathcal{H}_1}(A^c) \tag{17}$$

$$= 1 + q_{\mathcal{H}_0}(A) - q_{\mathcal{H}_1}(A) \tag{18}$$

$$\geq 1 + \inf_A (q_{\mathcal{H}_0}(A) - q_{\mathcal{H}_1}(A)) \tag{19}$$

$$= 1 - \sup_A (q_{\mathcal{H}_1}(A) - q_{\mathcal{H}_0}(A)) \tag{20}$$

$$= 1 - \delta_{TV}(q_{\mathcal{H}_0}, q_{\mathcal{H}_1}) \tag{21}$$

$\square$

## D. Implementation details

**GCIT**

In all our experiments we have set the depth of the generator, the discriminator and information network to 3. The number of hidden nodes in each layer is $d/10$ and $d/16$ for the generator and discriminator respectively ($d$ the number of inputs). For the information network, we use 2 diagonal matrices for each layer to make two hidden nodes for each feature separately. We use ReLu and tanh as the activation functions for each layer except for the output layer where we use a linear activation function for the information network, and sigmoid activation function for the discriminator and generator network given that we require its output to be constrained in the $(0, 1)$ interval and re-scale the data in the $(0, 1)$ interval prior to training. The number of samples in each mini-batch is 128 for the synthetic experiments and 64 for the genetic experiment. The GCIT and all experiments have been implemented and carried out in tensorflow and python. Pseudocode for the GCIT is given in Algorithm 1 and a python implementation is given at `https://github.com/alexisbellot/GCIT`.

---

**Algorithm 1** GCIT

---

**Input:** batch size $n_b$, data $\mathcal{D} = (\mathbf{x}, \mathbf{y}, \mathbf{z})$ of size $N$, statistic $\rho$, iterations $M$, parameter $\lambda$
**Initialize:** neural network model parameters $\phi, \eta, \theta$
**while** convergence criteria not satisfied **do**

    1. **Update Discriminator**
    Sample $\mathbf{z}_1, ..., \mathbf{z}_{n_b}$ from $\mathcal{D}$ and $\mathbf{v}_1, ..., \mathbf{v}_{n_b} \sim p_v$ a batch from the real and latent samples
    $\tilde{\mathbf{x}}_i \leftarrow G_\phi(\mathbf{z}_i, \mathbf{v}_i)$ for $i = 1, ..., n_b$
    Update $\eta$ by stochastic gradient descent with,

$$\nabla_\eta \frac{1}{n_b} \sum_{i=1}^{n_b} D_\eta(\mathbf{x}_i, \mathbf{z}_i) + (1 - D_\eta(\tilde{\mathbf{x}}_i, \mathbf{z}_i))$$

    2. **Update Information Network**
    Sample $\mathbf{z}_1, ..., \mathbf{z}_{n_b}$ from $\mathcal{D}$, $\mathbf{v}_1, ..., \mathbf{v}_{n_b} \sim p_v$ and $\kappa$ a permutation of $1, ..., n_b$
    $\tilde{\mathbf{x}}_i \leftarrow G_\phi(\mathbf{z}_i, \mathbf{v}_i)$ for $i = 1, ..., n_b$
    Update $\theta$ by stochastic gradient ascent with,

$$\nabla_\theta \left( \frac{1}{n_b} \sum_{i=1}^{n_b} T_\theta(\tilde{\mathbf{x}}_i, \mathbf{x}_i) - \log[\frac{1}{n_b} \sum_{i=1}^{n_b} \exp(T_\theta(\tilde{\mathbf{x}}_i, \mathbf{x}_{\kappa(i)}))] \right)$$

    3. **Update Generator**
    Sample $\mathbf{z}_1, ..., \mathbf{z}_{n_b}$ from $\mathcal{D}$ and $\mathbf{v}_1, ..., \mathbf{v}_{n_b} \sim p_v$
    $\tilde{\mathbf{x}}_i \leftarrow G_\phi(\mathbf{z}_i, \mathbf{v}_i)$ for $i = 1, ..., n_b$
    Update $\phi$ by stochastic gradient descent with,

$$\nabla_\phi(\mathcal{L}_G(D) + \lambda \mathcal{L}_{Info.})$$

**end while**
**for** $m = 1, ..., M$ **do**
    Sample $v_1, ..., v_N \sim p_v$
    $\tilde{x}_j^{(m)} \leftarrow G_\phi(z_j, v_j)$ for $j = 1, ..., N$
    $\hat{\rho}^{(m)} \leftarrow \rho(\tilde{\mathbf{x}}^{(m)}, \mathbf{y}, \mathbf{z})$
**end for**
$\hat{\rho} \leftarrow \rho(\mathbf{x}, \mathbf{y}, \mathbf{z})$
$\hat{p} \leftarrow \sum_{m=1}^M \mathbf{1}\{\hat{\rho}^{(m)} \geq \hat{\rho}\}/M$
**Output:** $p$-value $\hat{p}$

---

**Baseline algorithms**

We implemented the KCIT and RCoT with code provided by the authors in [11] in their R package `RCIT`, available at `https://github.com/ericstrobl/RCIT`. The CCIT [10] was implemented with the code provided by the authors at

https://github.com/rajatsen91/CCIT/blob/master/CCIT. The CRT was implemented in python with our own code.

# E: Genomics experiment details

The Cancer Cell Line Encyclopedia (CCLE) is a compilation of gene expression, chromosomal copy number and sequencing data from $947$ human cancer cell lines. A cancer cell line can be understood as a string of cancer cells that keep dividing and growing over time under certain conditions in a laboratory. Then, using high-throughput sequencing technologies, the molecular characteristics of cancer cell lines, such as gene expression or mutation data, can be extracted. These genetic predictors were coupled with measures of drug sensitivity for $PLX4720$: a drug used against cancer whose response is available for $474$ of the above cancer cell lines. By correlating the genetic information with the corresponding sensitivity to drug response, the data in principle allows for the identification of relevant genetic markers which could then lead to personalized treatment therapies depending on a patients genetic makeup. We illustrate this procedure in Figure 5.

Except for the conditional independence test to report significant genetic variables, our experiments followed similar procedures to those detailed in [12] and [1]. We choose to analyze dependence of drug response with $466$ genetic mutations observed on each cancer line. We give summary statistics of the final data used in Table 1 below. This is a very high-dimensional problem that makes conditional independence testing unfeasible with traditional tests.

As in the original study in [1], we proceeded by fitting an elastic net model to predict drug response from genetic features with $10$-fold cross-validation to optimize hyperparameters. Influential features were then ranked by their heuristic importance score given by the magnitude of fitted parameter values. The random forest model was used with default hyper-parameters in the python library `sklearn` and the CRT was implemented with a Gaussian approximation like in all other experiments. We ran the GCIT and the CRT considering each feature separately with drug response and all remaining features as confounders.

*Remark.* For a more systematic biological evaluation of features reported by the GCIT, we would use a more principled feature selection procedure such as Benjamini-Hochberg's correction for false discoveries [2].

Figure 5: Diagram illustrating the data used in the Genetic experiment.

Table 1: Summary statistics of the final genetic data used from [1].

| Statistics | Values |
|---|---|
| No. of cancer cell lines | 474 |
| No. of genetic mutations | 466 |
| Pearson correlation with drug response | **min:** $0.05$, **max:** $0.51$, **mean:** $0.07$, **var:** $0.001$ |
| Drug response distribution | **min:** $-97.9$, **max:** $43.3$, **mean:** $-17.2$, **var:** $633.3$ |