[Reviews · NeurIPS 2019]

Reviewer 1



Originality This paper presents a new way to use GANs in hypothesis testing. It was very interesting to use GANs to construct a null distribution that adapts to the dataset without strong assumptions. The proposed method can be used for feature selection and explainable neural networks. Quality Authors support their framework with theoretical justification and empirical results. The quantitative experimental results are limited to synthetic data but comprehensive and expected behaviours of the GCIT are observed in synthetic experiments. The only experimental result with real data is shown but it is hard to tell which result is more accurate and powerful. This issue will be raised when GCIT is applied to real-world applications. Clarity The method is clearly described and sufficient theoretical analysis was done. Exchangeability of samples and statistics were checked. The paper is well-written so that any machine learning scientists appreciate the main contribution and their intuition. Significance The main concern of this work boils down to the robustness of the method. As the authors have shown, the method dependent on hyperparameters (e.g., Lambda, the architecture of GANs) and quality of parameters of neural networks. Especially, in some academic fields, getting a p-value less than 0.05 is crucial to get the paper published. In this case, it is doubtful that the proposed method can be accepted in that community since by training models with bigger lambda or different GAN architectures will allow them to boost their p-values. The protocols to avoid overfitting of GANs, and choose hyperparameters and models should be carefully analyzed.

Reviewer 2



The authors tackle the problem of conditional independence testing problem by using a generative model to obtain the p-value. Pros: - Using GAN to generate conditional independent sample is a new design and a new method to tackle the problem. - Good result on control of the type 1 error - Promising simulation result Cons - Any rationale behind the reason why increasing lambda causes the increasing of type 1 error? - I feel that the sample size and scalability would be an issue here. If the data is limited would GAN be able to generate high quality samples of X? Also for a decent result how many sample should be generated until the p value become stable? It would be good if one investigate these problems for large implementations. Thank the authors for the feedback addressing my questions. I am keeping my score at 6.

Reviewer 3



This paper follows the framework of conditional independence tests provided by [4]. Instead of assuming the type of distribution, it uses GAN to simulate the distribution and then make conclusions based on distribution comparison. Some theoretic and empirical analysis of this method is provided in a logical manner. The experiments are well organized and easy to read. The framework is not new but introducing GAN may help to extend the testing to high dimensional data. Theoretic analysis does not completely cover all aspects that a hypothesis testing method needs.

[Author Response · NeurIPS 2019]



**We thank all reviewers for their thorough assessment of our paper.**

Figure 1: **Leftmost and middle-left panel**: Type I error and power as a function of sample size for data generated under scenario (3) with dimensionality of $Z$ set to 100; **Middle-right panel**: Empirical $p$-value variance of the GCIT as a function of sample size (computed by generating 100 $p$-values for each GAN trained on data with the specified size); **Rightmost panel**: Illustration of the convergence of the GCIT's $p$-values as a function of generated samples.

**Response to Reviewer #1.**

● *On the robustness of the GCIT for practical applications* - Indeed, our test does depend to some extent on the hyperparameter configurations. However, note that this dependence also exists in alternative tests such as the KCIT and RCoT (e.g. see Figure 1 in [1]), and the CCIT (given that it uses "parametrizable" classifiers). Recall that no ground truth is available to optimize hyperparameters using conventional methods, but we argue that the following procedure can be used to guide hyperparameter selection. We consider artificially inducing conditional independence $(X \perp\!\!\!\perp Y | Z)$ by permuting variables $X$ and $Y$ such as to preserve the marginal dependence in $(X, Z)$ and $(Y, Z)$, as in [2] (further details are also described in our related work). On this data, a well calibrated test is expected to produce uniformly distributed $p$-values, i.e. the empirical distribution of $p$-values should be approximately uniform. Our recommendation would be to choose GCIT's hyperparameters with lowest Kolmogorov-Smirnov statistic in comparison to the uniform distribution. This ensures the resulting test produces "well-behaved" $p$-values and thus prevents to some extent $p$-value cheating. We will discuss this further in the revised manuscript, thank you for raising this point.

**Response to Reviewer #3.**

● *On increasing Type I error with $\lambda$* - $\lambda$ determines the influence of $\mathcal{L}_{info}$ in the optimization of the generator (eq. 8). We do discuss the trade-off between power and type I error from a more qualitative, and perhaps intuitive, perspective in Section 3.3. However, insights can also be derived by considering the bound in Theorem 1. Theorem 1 shows that optimal control of the type I error is achieved by optimizing for $\mathcal{L}_G$ in isolation, i.e. $\lambda = 0$. Then, for $\lambda \neq 0$, optimizing for the additional $\mathcal{L}_{info}$ term may converge in practice to a higher $\mathcal{L}_G$, resulting in a higher upper-bound on type I error.

● *On the quality of generated samples and stability of p-values* - We investigate the influence of sample size on the three leftmost panels of Figure 1. The GCIT, as well as most competing tests, have slightly higher type I error in low sample sizes but control type I error successfully with 500 samples or more. In terms of power, our experiments show that we can expect the GCIT to outperform competing tests with 500 samples or more (for dimension of $Z = 100$). Next, we investigate the stability of $p$-values as a function of sample size; the variance of the empirical $p$-values quickly drops to 0. This means that for say 500 samples, we can expect the $p$-values of two independently trained GCITs to be within 0.005 of each other with approximately 95% confidence. The last panel on the right illustrates how quickly the $p$-value approximation (eq. 3) converges to its population quantity as a function of the number of samples used to compute the approximation i.e. $M$ in eq. 3. The convergence should be at least of order $M^{-1/2}$ by the central limit theorem. As an alternative to a default number of generated samples (previously $M = 1000$), this last experiment led us to modify our implementation to stop sampling from the trained GAN whenever the computed $p$-value is within $1e^{-3}$ of the mean of the previous 100 computed $p$-values. This has reduced the computational complexity of the overall procedure while giving better or similar performance, thank you for the suggestion.

**Response to Reviewer #4.**

● *On the use of GANs as generative models* - From a practical perspective, algorithms based on other generative models can be constructed based on our proposed procedure. However, we chose GANs as they have analytical properties that allow deriving the error bounds in Theorem 1 and enable us to maximize power explicitly with the addition of the information network. In practice, good performance may also be achieved using other flexible generative models. We will mention this as interesting future research.

● *On the quality of generated samples and stability of p-values* - Please kindly refer to the response to Reviewer #3.

[1] Zhang, Kun, et al. "Kernel-based conditional independence test and application in causal discovery." UAI. 2012.

[2] Doran, Gary, et al. "A Permutation-Based Kernel Conditional Independence Test." UAI. 2014.


[Meta-Review · NeurIPS 2019]

The paper proposes a novel and interesting GAN-based method for conditional independence test. While there is some concern on applying the black-box method to rigorous statistical tests, I still think the paper includes a new and significant idea to the important but difficult problem of conditional independence. I would like the authors to reflect the review comments in making the final version.